# EFFICIENT SHARPNESS-AWARE MINIMIZATION FOR IMPROVED TRAINING OF NEURAL NETWORKS

**Jiawei Du**[1,2] , **Hanshu Yan**[2] , **Jiashi Feng**[2] , **Joey Tianyi Zhou**[1,*] **Liangli Zhen**[4] ,
**Rick Siow Mong Goh**[4] , **Vincent Y. F. Tan**[3,2]

[1]Centre for Frontier AI Research (CFAR), A*STAR, Singapore,
[2]Department of Electrical and Computer Engineering, National University of Singapore
[3]Department of Mathematics, National University of Singapore
[4]Institute of High Performance Computing (IHPC), A*STAR, Singapore
`{dujiawei, hanshu.yan}@u.nus.edu,vtan@nus.edu.sg`
`jshfeng@gmail.com, Joey.tianyi.zhou@gmail.com`
`{zhen_liangli, gohsm}@ihpc.a-star.edu.sg`

## ABSTRACT

Overparametrized Deep Neural Networks (DNNs) often achieve astounding performances, but may potentially result in severe generalization error. Recently, the relation between the sharpness of the loss landscape and the generalization error has been established by Foret et al. (2020), in which the Sharpness Aware Minimizer (SAM) was proposed to mitigate the degradation of the generalization. Unfortunately, SAM's computational cost is roughly double that of base optimizers, such as Stochastic Gradient Descent (SGD). This paper thus proposes *Efficient Sharpness Aware Minimizer* (ESAM), which boosts SAM's efficiency at no cost to its generalization performance. ESAM includes two novel and efficient training strategies—Stochastic Weight Perturbation and Sharpness-Sensitive Data Selection. In the former, the sharpness measure is approximated by perturbing a stochastically chosen set of weights in each iteration; in the latter, the SAM loss is optimized using only a judiciously selected subset of data that is sensitive to the sharpness. We provide theoretical explanations as to why these strategies perform well. We also show, via extensive experiments on the CIFAR and ImageNet datasets, that ESAM enhances the efficiency over SAM from requiring $100\%$ extra computational overhead to $40\%$ vis-à-vis base optimizers, while test accuracies are preserved or even improved. Our codes are avaliable at https://github.com/dydjw9/Efficient_SAM.

## 1 INTRODUCTION

Deep learning has achieved astounding performances in many fields by relying on larger numbers of parameters and increasingly sophisticated optimization algorithms. However, DNNs with far more parameters than training samples are more prone to poor generalization. Generalization is arguably the most fundamental and yet mysterious aspect of deep learning.

Several studies have been conducted to better understand the generalization of DNNs and to train DNNs that generalize well across the natural distribution (Keskar et al., 2017; Neyshabur et al., 2017; Chaudhari et al., 2019; Zhang et al., 2019; Wu et al., 2020; Foret et al., 2020; Zhang et al., 2021). For example, Keskar et al. (2017) investigate the effect of batch size on neural networks' generalization ability. Zhang et al. (2019); Zhou et al. (2021) propose optimizers for training DNNs with improved generalization ability. Specifically, Hochreiter & Schmidhuber (1995), Li et al. (2018) and Dinh et al. (2017) argue that the geometry of the loss landscape affects generalization and DNNs with a flat minimum can generalize better. The recent work by Foret et al. (2020) proposes an effective training algorithm *Sharpness Aware Minimizer (SAM)* for obtaining a flat minimum. SAM employs a base optimizer such as Stochastic Gradient Descent (Nesterov, 1983) or Adam (Kingma & Ba, 2015) to minimize both the vanilla training loss and the sharpness. The sharpness, which describes

---
*corresponding author

the flatness of a minimum, is characterized using eigenvalues of the Hessian matrix by Keskar et al. (2017). SAM quantifies the sharpness as the maximized change of training loss when a constraint perturbation is added to current weights. As a result, SAM leads to a flat minimum and significantly improves the generalization ability of the trained DNNs. SAM and its variants have been shown to outperform the state-of-the-art across a variety of deep learning benchmarks (Kwon et al., 2021; Chen et al., 2021; Galatolo et al., 2021; Zheng et al., 2021). Regrettably though, SAM and its variants achieve such remarkable performance at the expense of doubling the computational overhead of the given base optimizers, which minimize the training loss with a single forward and backward propagation step. SAM requires an additional propagation step compared to the base optimizers to resolve the weight perturbation for quantifying the sharpness. The extra propagation step requires the same computational overhead as the single propagation step used by base optimizers, resulting in SAM's computational overhead being *doubled* ($2\times$). As demonstrated in Figure 1, SAM achieves higher test accuracy (i.e., $84.46\%$ vs. $81.89\%$) at the expense of sacrificing half of the training speed of the base optimizer (i.e., 276 imgs/s vs. 557 imgs/s).

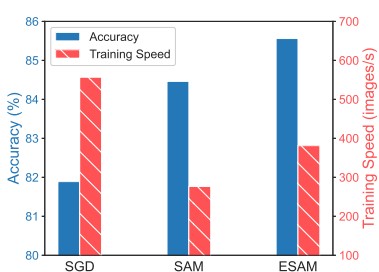

Figure 1: Training Speed vs. Accuracy of SGD, SAM and ESAM evaluated by PyramidNet on CIFAR100. ESAM improves the efficiency with better accuracy compared to SAM.

In this paper, we aim to improve the efficiency of SAM but preserve its superior performance in generalization. We propose *Efficient Sharpness Aware Minimizer (ESAM)*, which consists of two training strategies *Stochastic Weight Perturbation (SWP)* and *Sharpness-sensitive Data Selection (SDS)*, both of which reduce computational overhead and preserve the performance of SAM. On the one hand, SWP approximates the sharpness by searching weight perturbation within a stochastically chosen neighborhood of the current weights. SWP preserves the performance by ensuring that the expected weight perturbation is identical to that solved by SAM. On the other hand, SDS improves efficiency by approximately optimizing weights based on the sharpness-sensitive subsets of batches. These subsets consist of samples whose loss values increase most w.r.t. the weight perturbation and consequently can better quantify the sharpness of DNNs. As a result, the sharpness calculated over the subsets can serve as an upper bound of the SAM's sharpness, ensuring that SDS's performance is comparable to that of SAM's.

We verify the effectiveness of ESAM on the CIFAR10, CIFAR100 (Krizhevsky et al., 2009) and ImageNet (Deng et al., 2009) datasets with five different DNN architectures. The experimental results demonstrate that ESAM obtains flat minima at a cost of only $40\%$ (vs. SAM's $100\%$) extra computational overhead over base optimizers. More importantly, ESAM achieves better performance in terms of the test accuracy compared to SAM. In a nutshell, our contributions are as follows:

- We propose two novel and effective training strategies *Stochastic Weight Perturbation (SWP)* and *Sharpness-sensitive Data Selection (SDS)*. Both strategies are designed to improve efficiency without sacrificing performance. The empirical results demonstrate that both of the proposed strategies can improve both the efficiency and effectiveness of SAM.

- We introduce the ESAM, which integrates SWP and SDS. ESAM improves the generalization ability of DNNs with marginally additional computational cost compared to standard training.

The rest of this paper is structured in this way. Section 2.1 introduces SAM and its computational issues. Section 2.2 and Section 2.3 discuss how the two proposed training strategies SWP and SDS are designed respectively. Section 3 verifies the effectiveness of ESAM across a variety of datasets and DNN architectures. Section 4 presents the related work and Section 5 concludes this paper.

## 2 METHODOLOGY

We start with recapitulating how SAM achieves a flat minimum with small sharpness, which is quantified by resolving a maximization problem. To compute the sharpness, SAM requires additional forward and backward propagation and results in the doubling of the computational overhead

---

**Algorithm 1** Efficient SAM (ESAM)

---

**Input:** Network $f_\theta$ with parameters $\theta = (\theta_1, \theta_2, \ldots, \theta_N)$; Training set $\mathbb{S}$; Batch size $b$; Learning rate $\eta > 0$; Neighborhood size $\rho > 0$; Number of iterations $A$; SWP hyperparameter $\beta$; SDS hyperparameter $\gamma$.

**Output:** A flat minimum solution $\hat{\theta}$.

1: **for** $a = 1$ to $A$ **do**
2:     Sample a mini-batch $\mathbb{B} \subset \mathbb{S}$ with size $b$.
3:     **for** $n = 1$ to $N$ **do**
4:         **if** $\theta_n$ is chosen by probability $\beta$ **then**
5:             $\epsilon_n \leftarrow \frac{\rho}{1-\beta}\nabla_{\theta_n}L_{\mathbb{B}}(f_\theta)$                                  ▷ SWP in $B_1$
6:         **else**
7:             $\epsilon_n \leftarrow 0$
8:         $\hat{\epsilon} \leftarrow (\epsilon_1, \ldots, \epsilon_N)$                           ▷ Assign Weight Perturbation
9:         Compute $\ell(f_{\theta+\hat{\epsilon}}, x_i, y_i)$ and construct $\mathbb{B}^+$ with selection ratio $\gamma$ (Equation 6)
10:       Compute gradients $g = \nabla_\theta L_{\mathbb{B}^+}(f_{\theta+\hat{\epsilon}})$                 ▷ SDS in $B_2$
11:       Update weights $\theta \leftarrow \theta - \eta g$

---

compared to base optimizers. Following that, we demonstrate how we derive and propose ESAM, which integrates SWP and SDS, to maximize efficiency while maintaining the performance. We introduce SWP and SDS in Sections 2.2 and 2.3 respectively. Algorithm 1 shows the overall proposed ESAM algorithm.

Throughout this paper, we denote a neural network $f$ with weight parameters $\theta$ as $f_\theta$. The weights are contained in the vector $\theta = (\theta_1, \theta_2, \ldots, \theta_N)$, where $N$ is the number of weight units in the neural network. Given a training dataset $\mathbb{S}$ that contains samples i.i.d. drawn from a distribution $\mathcal{D}$, the network is trained to obtain optimal weights $\hat{\theta}$ via *empirical risk minimization* (ERM), i.e.,

$$\hat{\theta} = \arg\min_\theta \left\{ L_{\mathbb{S}}(f_\theta) = \frac{1}{|\mathbb{S}|} \sum_{(x_i, y_i) \in \mathbb{S}} \ell(f_\theta, x_i, y_i) \right\}. \tag{1}$$

where $\ell$ can be an arbitrary loss function. We take $\ell$ to be the cross entropy loss in this paper. The *population loss* is defined as $L_{\mathcal{D}}(f_\theta) \triangleq \mathbb{E}_{(x_i, y_i) \sim \mathcal{D}}[\ell(f_\theta, x_i, y_i)]$. In each training iteration, optimizers sample a mini-batch $\mathbb{B} \subset \mathbb{S}$ with size $b$ to update parameters.

## 2.1 SHARPNESS-AWARE MINIMIZATION AND ITS COMPUTATIONAL DRAWBACK

To improve the generalization capability of DNNs, Foret et al. (2020) proposed the SAM training strategy for searching flat minima. SAM trains DNNs by solving the following min-max optimization problem,

$$\min_\theta \max_{\epsilon:\|\epsilon\|_2 \le \rho} L_{\mathbb{S}}(f_{\theta+\epsilon}). \tag{2}$$

Given $\theta$, the inner optimization attempts to find a weight perturbation $\epsilon$ in Euclidean ball with radius $\rho$ that maximizes the empirical loss. The maximized loss at weights $\theta$ is the sum of the empirical loss and the sharpness, which is defined to be $R_{\mathbb{S}}(f_\theta) = \max_{\epsilon:\|\epsilon\|_2 < \rho}[L_{\mathbb{S}}(f_{\theta+\epsilon}) - L_{\mathbb{S}}(f_\theta)]$. This sharpness is quantified by the maximal change of empirical loss when a perturbation $\epsilon$ (whose norm is constrained by $\rho$) is added to $\theta$. The min-max problem encourages SAM to find flat minima.

For a certain set of weights $\theta$, Foret et al. (2020) theoretically justifies that the population loss of DNNs can be upper-bounded by the sum of sharpness, empirical loss, and a regularization term on the norm of weights (refer to Equation 3). Thus, by minimizing the sharpness together with the empirical loss, SAM produces optimized solutions for DNNs with flat minima, and the resultant models can thus generalize better (Foret et al., 2020; Chen et al., 2021; Kwon et al., 2021). Indeed, we have

$$L_{\mathcal{D}}(f_\theta) \le R_{\mathbb{S}}(f_\theta) + L_{\mathbb{S}}(f_\theta) + \lambda\|\theta\|_2^2 = \max_{\epsilon:\|\epsilon\|_2 \le \rho} L_{\mathbb{S}}(f_{\theta+\epsilon}) + \lambda\|\theta\|_2^2. \tag{3}$$

In practice, SAM first approximately solves the inner optimization by means of a single-step gradient descent method, i.e.,

$$\hat{\epsilon} = \arg\max_{\epsilon:\|\epsilon\|_2 < \rho} L_{\mathbb{S}}(f_{\theta+\epsilon}) \approx \rho\nabla_\theta L_{\mathbb{S}}(f_\theta). \tag{4}$$

The sharpness at weights $\theta$ is approximated by $R_{\mathbb{S}}(f_\theta) = L_{\mathbb{S}}(f_{\theta+\hat{\epsilon}}) - L_{\mathbb{S}}(f_\theta)$. Then, a base optimizer, such as SGD (Nesterov, 1983) or Adam (Kingma & Ba, 2015), updates the DNNs' weights to minimize $L_{\mathbb{S}}(f_{\theta+\hat{\epsilon}})$. We refer to $L_{\mathbb{S}}(f_{\theta+\hat{\epsilon}})$ as the *SAM loss*. Overall, SAM requires two forward and two backward operations to update weights once. We refer to the forward and backward propagation for approximating $\hat{\epsilon}$ as $F_1$ and $B_1$ and those for updating weights by base optimizers as $F_2$ and $B_2$ respectively. Although SAM can effectively improve the generalization of DNNs, it additionally requires one forward and one backward operation ($F_1$ and $B_1$) in each training iteration. Thus, SAM results in a *doubling* of the computational overhead compared to the use of base optimizers.

To improve the efficiency of SAM, we propose ESAM, which consists of two strategies—SWP and SDS, to accelerate the sharpness approximation phase and the weight updating phase. Specifically, on the one hand, when estimating $\hat{\epsilon}$ around weight vector $\theta$, SWP efficiently approximates $\hat{\epsilon}$ by randomly selecting each parameter with a given probability to form a subset of weights to be perturbed. The reduction of the number of perturbed parameters results in lower computational overhead during the backward propagation. SWP rescales the resultant weight perturbation so as to assure that the expected weight perturbation equals to $\hat{\epsilon}$, and the generalization capability thus will not be significantly degraded. On the other hand, when updating weights via base optimizers, instead of computing the upper bound $L_{\mathbb{B}}(f_{\theta+\hat{\epsilon}})$ over a whole batch of samples, SDS selects a *subset* of samples, $\mathbb{B}^+$, whose loss values increase the most with respect to the perturbation $\hat{\epsilon}$. Optimizing the weights based on a fewer number of samples decreases the computational overhead (in a linear fashion). We further justify that $L_{\mathbb{B}}(f_{\theta+\hat{\epsilon}})$ can be upper bounded by $L_{\mathbb{B}^+}(f_{\theta+\hat{\epsilon}})$ and consequently the generalization capability can be preserved. In general, ESAM works much more efficiently and performs as well as SAM in terms of the generalization capability.

## 2.2 Stochastic Weight Perturbation

This section elaborates on the first efficiency enhancement strategy, SWP, and explains why SWP can effectively reduce computational overhead while preserving the generalization capability.

To efficiently approximate $\hat{\epsilon}(\theta, \mathbb{S})$ during the sharpness estimation phase, SWP randomly chooses a subset $\tilde{\theta} = \{\theta_{I_1}, \theta_{I_2}, \ldots\}$ from the original set of weights $\theta = (\theta_1, \ldots, \theta_N)$ to perform backpropagation $B_1$. Each parameter is selected to be in the subvector $\tilde{\theta}$ with some probability $\beta$, which can be tuned as a hyperparameter. SWP approximates the weight perturbation with $\rho\nabla_{\tilde{\theta}}L_{\mathbb{S}}(f_\theta)$. To be formal, we introduce a *gradient mask* $\mathbf{m} = (m_1, \ldots, m_N)$ where $m_i \overset{\text{i.i.d.}}{\sim} \text{Bern}(\beta)$ for all $i \in \{1, \ldots, N\}$. Then, we have $\rho\nabla_{\tilde{\theta}}L_{\mathbb{S}}(f_\theta) = \mathbf{m}^\top\hat{\epsilon}(\theta, \mathbb{B})$. To ensure the expected weight perturbation of SWP equals to $\hat{\epsilon}$, we scale $\rho\nabla_{\tilde{\theta}}L_{\mathbb{S}}(f_\theta)$ by a factor of $\frac{1}{\beta}$. Finally, SWP produces an approximate solution of the inner maximization as

$$\boldsymbol{a}(\theta, \mathbb{B}) = \frac{\mathbf{m}^\top\hat{\epsilon}(\theta, \mathbb{B})}{\beta}. \tag{5}$$

**Computation** Ideally, SWP reduces the overall computational overhead in proportion to $1 - \beta$ in $B_1$. However, there exists some parameters not included in $\tilde{\theta}$ that are still required to be updated in the backpropagation step. This additional computational overhead is present due to the use of the chain rule, which calculates the entire set of gradients with respect to the parameters along a propagation path. This additional computational overhead slightly increases in deeper neural networks. Thus, the amount of reduction in the computational overhead is positively correlated to $1 - \beta$. In practice, $\beta$ is tuned to maximize SWP's efficiency while maintaining a generalization performance comparable to SAM's.

**Generalization** We will next argue that SWP's generalization performance can be preserved when compared to SAM by showing that the expected weight perturbation $\boldsymbol{a}(\theta, \mathbb{B})$ of SWP equals to the original SAM's perturbation $\hat{\epsilon}(\theta, \mathbb{B})$ in the sense of the $\ell_2$ norm and direction. We denote the expected SWP perturbation by $\bar{\boldsymbol{a}}(\theta, \mathbb{B})$, where

$$\bar{\boldsymbol{a}}(\theta, \mathbb{B})_{[i]} = \mathbb{E}[\boldsymbol{a}(\theta, \mathbb{B})_{[i]}] = \frac{1}{\beta} \cdot \beta\hat{\epsilon}(\theta, \mathbb{B})_{[i]} = \hat{\epsilon}(\theta, \mathbb{B})_{[i]},$$

for $i \in \{1, \ldots, N\}$. Thus, it holds that

$$\|\bar{\boldsymbol{a}}(\theta, \mathbb{B})\|_2 = \|\hat{\epsilon}(\theta, \mathbb{B})\|_2 \quad \text{and} \quad \text{CosSim}\Big(\bar{\boldsymbol{a}}(\theta, \mathbb{B}), \hat{\epsilon}(\theta, \mathbb{B})\Big) = 1,$$

showing that the expected weight perturbation of SWP is the same as that of SAM's.

## 2.3 Sharpness-sensitive Data Selection

In this section, we introduce the second efficiency enhancement technique, SDS, which reduces computational overhead of SAM linearly as the number of selected samples decreases. We also explain why the generalization capability of SAM is preserved by SDS.

In the sharpness estimation phase, we obtain the approximate solution $\hat{\epsilon}$ of the inner maximization. Perturbing weights along this direction significantly increases the average loss over a batch $\mathbb{B}$. To improve the efficiency but still control the upper bound $L_{\mathbb{B}}(f_{\theta+\hat{\epsilon}})$, we select a subset of samples from the whole batch. The loss values of this subset of samples increase most when the weights are perturbed by $\hat{\epsilon}$. To be specific, SDS splits the mini-batch $\mathbb{B}$ into the following two subsets

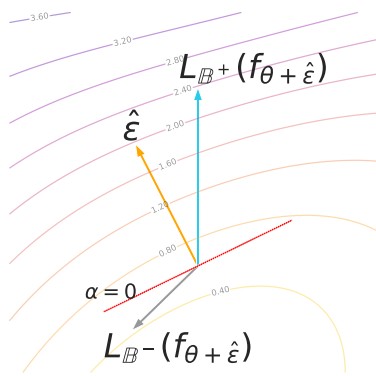

Figure 2: Illustration on the loss changes of samples in $\mathbb{B}^+$ and $\mathbb{B}^-$ along the weight perturbation $\hat{\epsilon}$. The average loss of samples in $\mathbb{B}^+$ increases the most along the perturbation direction $\hat{\epsilon}$.

$$\begin{aligned} \mathbb{B}^+ &:= \big\{ (x_i, y_i) \in \mathbb{B} : \ell(f_{\theta+\hat{\epsilon}}, x_i, y_i) - \ell(f_\theta, x_i, y_i) > \alpha \big\}, \\ \mathbb{B}^- &:= \big\{ (x_i, y_i) \in \mathbb{B} : \ell(f_{\theta+\hat{\epsilon}}, x_i, y_i) - \ell(f_\theta, x_i, y_i) < \alpha \big\}, \end{aligned} \quad (6)$$

where $\mathbb{B}^+$ is termed as the sharpness-sensitive subset and the threshold $\alpha$ controls the size of $\mathbb{B}^+$. We let $\gamma = |\mathbb{B}^+|/|\mathbb{B}|$ be the ratio of the number of selected samples with respect to the batch size. In practice, $\gamma$ determines the exact value of $\alpha$ and serves as a predefined hyperparameter of SDS. As illustrated in Figure 2, when $\alpha = 0$, the gradient of the weights evaluated on $\mathbb{B}^+$ *aligns* with the direction of $\hat{\epsilon}$ and the loss values of the samples in $\mathbb{B}^+$ will increase with respect to the weight perturbation $\hat{\epsilon}$.

**Computation** SDS reduces the computational overhead in $F_2$ and $B_2$. The reduction is linear in $1 - \gamma$. The hyperparameter $\gamma$ can be tuned to meet up distinct requirements in efficiency and performance. SDS is configured the same as SWP for maximizing efficiency with comparable performance to SAM.

**Generalization** For the generalization capability, we now justify that the SAM loss computing over the batch $\mathbb{B}$, $L_{\mathbb{B}}(f_{\theta+\hat{\epsilon}})$, can be approximately upper bounded by the corresponding loss evaluated only on $\mathbb{B}^+$, $L_{\mathbb{B}^+}(f_{\theta+\hat{\epsilon}})$. From Equation 3, we have

$$\begin{aligned} L_{\mathbb{B}}(f_{\theta+\hat{\epsilon}}) &= \gamma L_{\mathbb{B}^+}(f_{\theta+\hat{\epsilon}}) + (1-\gamma) L_{\mathbb{B}^-}(f_{\theta+\hat{\epsilon}}) \\ &= L_{\mathbb{B}^+}(f_{\theta+\hat{\epsilon}}) + (1-\gamma)[L_{\mathbb{B}^-}(f_{\theta+\hat{\epsilon}}) - L_{\mathbb{B}^+}(f_{\theta+\hat{\epsilon}})] \\ &= L_{\mathbb{B}^+}(f_{\theta+\hat{\epsilon}}) + (1-\gamma)[R_{\mathbb{B}^-}(f_\theta) + L_{\mathbb{B}^-}(f_\theta) - R_{\mathbb{B}^+}(f_\theta) - L_{\mathbb{B}^+}(f_\theta)]. \end{aligned} \quad (7)$$

On the one hand, since $R_{\mathbb{B}}(f_\theta) = \frac{1}{|\mathbb{B}|}\sum_{(x_i,y_i)\in\mathbb{B}}[\ell(f_{\theta+\hat{\epsilon}}, x_i, y_i) - \ell(f_\theta, x_i, y_i)]$ represents the average sharpness of the batch $\mathbb{B}$, by Equation 6, we have $R_{\mathbb{B}^-}(f_\theta) \le R_{\mathbb{B}}(f_\theta) \le R_{\mathbb{B}^+}(f_\theta)$, and

$$R_{\mathbb{B}^-}(f_\theta) - R_{\mathbb{B}^+}(f_\theta) \le 0. \quad (8)$$

On the other hand, $\mathbb{B}^+$ and $\mathbb{B}^-$ are constructed by sorting $\ell(f_{\theta+\hat{\epsilon}}, x_i, y_i) - \ell(f_\theta, x_i, y_i)$, which is positively correlated to $l(f_\theta, x_i, y_i)$ (Li et al., 2019) (more details can be found in Appendix A.2). Thus, we have

$$L_{\mathbb{B}^-}(f_\theta) - L_{\mathbb{B}^+}(f_\theta) \le 0. \quad (9)$$

Therefore, by Equation 8 and Equation 9, we have

$$L_{\mathbb{B}}(f_{\theta+\hat{\epsilon}}) \le L_{\mathbb{B}^+}(f_{\theta+\hat{\epsilon}}). \quad (10)$$

Experimental results in Figure 5 corroborate that $R_{\mathbb{B}^-}(f_\theta) - R_{\mathbb{B}^+}(f_\theta) < 0$ and $L_{\mathbb{B}^-}(f_\theta) - L_{\mathbb{B}^+}(f_\theta) < 0$. Besides, Figure 6 verifies that the selected batch $\mathbb{B}^+$ is sufficiently representative to mimic the gradients of $\mathbb{B}$ since $\mathbb{B}^+$ has a significantly higher cosine similarity with $\mathbb{B}$ compared to $\mathbb{B}^-$ in terms of the computed gradients. According to Equation 10, one can utilize $L_{\mathbb{B}^+}(f_{\theta+\hat{\epsilon}})$ as a proxy to the real objective to minimize of the overall loss $L_{\mathbb{B}}(f_{\theta+\hat{\epsilon}})$ with a smaller number of samples. As a result, SDS improves SAM's efficiency without performance degradation.

## 3 EXPERIMENTS

This section demonstrates the effectiveness of our proposed ESAM algorithm. We conduct experiments on several benchmark datasets: CIFAR-10 (Krizhevsky et al., 2009), CIFAR-100 (Krizhevsky et al., 2009) and ImageNet (Deng et al., 2009), using various model architectures: ResNet (He et al., 2016), Wide ResNet (Zagoruyko & Komodakis, 2016), and PyramidNet (Han et al., 2017). We demonstrate the proposed ESAM improves the efficiency of vanilla SAM by speeding up to $40.3\%$ computational overhead with better generalization performance. We report the main results in Table 1 and Table 2. Besides, we perform an ablation study on the two proposed strategies of ESAM (i.e., SWP and SDS). The experimental results in Table 3 and Figure 3 indicate that both strategies improve SAM's efficiency and performance.

### 3.1 RESULTS

**CIFAR10 and CIFAR100** We start from evaluating ESAM on the CIFAR-10 and CIFAR-100 image classification datasets. The evaluation is carried out on three different model architectures: ResNet-18 (He et al., 2016), WideResNet-28-10 (Zagoruyko & Komodakis, 2016) and PyramidNet-110 (Han et al., 2017). We set all the training settings, including the maximum number of training epochs, iterations per epoch, and data augmentations, the same for fair comparison among SGD, SAM and ESAM. Additionally, the other hyperparameters of SGD, SAM and ESAM have been tuned separately for optimal test accuracies using grid search.

We train all the models with 3 different random seeds using a batch size of 128, weight decay $10^{-4}$ and cosine learning rate decay (Loshchilov & Hutter, 2017). The training epochs are set to be 200 for ResNet-18 (He et al., 2016), WideResNet-28-10 (Zagoruyko & Komodakis, 2016), and 300 for PyramidNet-110 (Han et al., 2017). We set $\beta = 0.6$ and $\gamma = 0.5$ for ResNet-18 and PyramidNet-110 models; and set $\beta = 0.5$ and $\gamma = 0.5$ for WideResNet-28-10. The above-mentioned $\beta$ and $\gamma$ are optimal for efficiency with comparable performance compared to SAM. The details of training setting are listed in Appendix A.7. We record the best test accuracies obtained by SGD, SAM and ESAM in Table 1.

The experimental results indicate that our proposed ESAM can increase the training speed by up to $40.30\%$ in comparison with SAM. Concerning the performance, ESAM outperforms SAM in the six sets of experiments. The best efficiency of ESAM is reported in CIFAR10 trained with ResNet-18 (Training speed $140.3\%$ vs. SAM $100\%$). The best accuracy is reported in CIFAR100 trained with PyramidNet110 (Accuracy $85.56\%$ vs. SAM $84.46\%$). ESAM improves efficiency and achieves better performance compared to SAM in CIFAR10/100 benchmarks.

Table 1: Classification accuracies and training speed on the CIFAR-10 and CIFAR-100 datasets. Computational overhead is quantified by #images processed per second (images/s). The numbers in parentheses ($\cdot$) indicate the ratio of ESAM's training speed w.r.t. SAM.

|  | **CIFAR-10** | | **CIFAR-100** | |
|---|---|---|---|---|
| **ResNet-18** | Accuracy | images/s | Accuracy | images/s |
| SGD | $95.41_{\pm 0.03}$ | 3,387 | $78.17_{\pm 0.05}$ | 3,483 |
| SAM | $96.52_{\pm 0.13}$ | 1,717(100.0%) | $80.17_{\pm 0.17}$ | 1,730 (100.0%) |
| ESAM | $\mathbf{96.56}_{\pm 0.08}$ | 2,409 (140.3%) | $\mathbf{80.41}_{\pm 0.10}$ | 2,423 (140.0%) |
| **Wide-28-10** | Accuracy | images/s | Accuracy | images/s |
| SGD | $96.34_{\pm 0.12}$ | 801 | $81.56_{\pm 0.13}$ | 792 |
| SAM | $97.27_{\pm 0.11}$ | 396 (100.0%) | $83.42_{\pm 0.04}$ | 391 (100.0%) |
| ESAM | $\mathbf{97.29}_{\pm 0.11}$ | 550 (138.9%) | $\mathbf{84.51}_{\pm 0.01}$ | 545 (139.4%) |
| **PyramidNet-110** | Accuracy | images/s | Accuracy | images/s |
| SGD | $96.62_{\pm 0.10}$ | 580 | $81.89_{\pm 0.17}$ | 555 |
| SAM | $97.30_{\pm 0.10}$ | 289 (100.0%) | $84.46_{\pm 0.04}$ | 276 (100.0%) |
| ESAM | $\mathbf{97.81}_{\pm 0.01}$ | 401 (138.7%) | $\mathbf{85.56}_{\pm 0.05}$ | 381 (137.9%) |

**ImageNet** To evaluate ESAM's effectiveness on a large-scale benchmark dataset, we conduct experiments on ImageNet Datasets. The 1000 class ImageNet dataset contains roughly 1.28 million

Table 2: Classification accuracies and training speed on the ImageNet dataset. The numbers in parentheses (·) indicate the ratio of ESAM's training speed w.r.t. SAM's. Results with * are referred to Chen et al. (2021)

| ImageNet | ResNet-50 | | ResNet-101 | |
|---|---|---|---|---|
| | Accuracy | images/s | Accuracy | images/s |
| SGD | 76.00* | 1,327 | 77.80* | 891 |
| SAM | 76.70* | 654 (100.0%) | 78.60* | 438 (100.0%) |
| ESAM | 77.05 | 846 (129.3%) | 79.09 | 564 (128.7%) |

training images and $50,000$ validation images with $469 \times 387$ averaged resolution. The ImageNet dataset is more representative (of real-world scenarios) and persuasive (of a method's effectiveness) than CIFAR datasets. We resize the images on ImageNet to $224 \times 224$ resolution to train ResNet-50 and ResNet-101 models. We train 90 epochs and set the optimal hyperparameters for SGD, SAM and ESAM as suggested by Chen et al. (2021), and the details are listed in appendix A.7. We use $\beta = 0.6$ and $\gamma = 0.7$ for ResNet-50 and ResNet-101. We employ the $m$-sharpness strategy for both SAM and ESAM with $m = 128$, which is the same as that suggested in Zheng et al. (2021).

The experimental results are reported in Table 2. The results indicate that the performance of ESAM on large-scale datasets is consistent with the two (smaller) CIFAR datasets. ESAM outperforms SAM by $0.35\%$ to $0.49\%$ in accuracy and, more importantly, enjoys $28.7\%$ faster training speed compared to SAM. As the $\gamma$ we used here is larger than the one used in CIFAR datasets, the training speed of ESAM here is slightly slower than that in the CIFAR datasets.

These experiments demonstrate that ESAM outperforms SAM on a variety of benchmark datasets for widely-used DNNs' architectures in terms of training speed and classification accuracies.

## 3.2 ABLATION AND PARAMETER STUDIES

To better understand the effectiveness of SWP and SDS in improving the performance and efficiency compared to SAM, we conduct four sets of ablation studies on CIFAR-10 and CIFAR-100 datasets using ResNet-18 and WideResNet-28-10 models, respectively. We consider two variants of ESAM: (i) only with SWP, (ii) only with SDS. The rest of the experimental settings are identical to the settings described in Section 3.1. We conduct grid search over the interval $[0.3, 0.9]$ for $\beta$ and the interval $[0.3, 0.9]$ for $\gamma$, with a same step size of $0.1$. We report the grid search results in Figure 3. We use $\beta = 0.6, \gamma = 0.5$ for ResNet-18; and set $\beta = 0.5$, $\gamma = 0.5$ for WideResNet-28-10 in the four sets of ablation studies. The ablation study results are reported in Table 3.

Table 3: Ablation Study of ESAM on CIFAR-10 and CIFAR100. The numbers in brackets [·] represent the accuracy improvement in comparison to SGD. The numbers in parentheses (·) indicate the ratio of ESAM's training speed to SAM's. Green color indicates improvement compared to SAM, whereas red color suggests a degradation.

| ResNet-18 | CIFAR-10 | | CIFAR-100 | |
|---|---|---|---|---|
| | Accuracy | images/s | Accuracy | images/s |
| SGD | 95.41 | 3,387 | 78.17 | 3,438 |
| SAM | 96.52 [+1.11] | 1,717 (100.0%) | 80.17 [+2.00] | 1,730 (100.0%) |
| + ESAM-SWP | **96.74** [+1.33] | 1,896 (110.5%) | **80.53** [+2.36] | 1,887 (109.1%) |
| + ESAM-SDS | 96.45 [+1.04] | 2,105 (122.6%) | 80.38 [+2.21] | 2,103 (121.5%) |
| ESAM | 96.56 [+1.15] | **2,409** (140.3%) | 80.41 [+2.24] | **2,423** (140.9%) |
| **Wide-28-10** | Accuracy | images/s | Accuracy | images/s |
| SGD | 96.34 | 801 | 81.56 | 792 |
| SAM | 97.27 [+0.93] | 396 (100.0%) | 83.42 [+1.86] | 391 (100.0%) |
| + ESAM-SWP | **97.37** [+1.03] | 430 (108.5%) | 84.44 [+2.88] | 423 (108.3%) |
| + ESAM-SDS | 97.24 [+0.90] | 495 (124.8%) | 84.46 [+2.90] | 492 (125.8%) |
| ESAM | 97.29 [+0.95] | **551** (138.9%) | **84.51** [+2.95] | **545** (139.4%) |

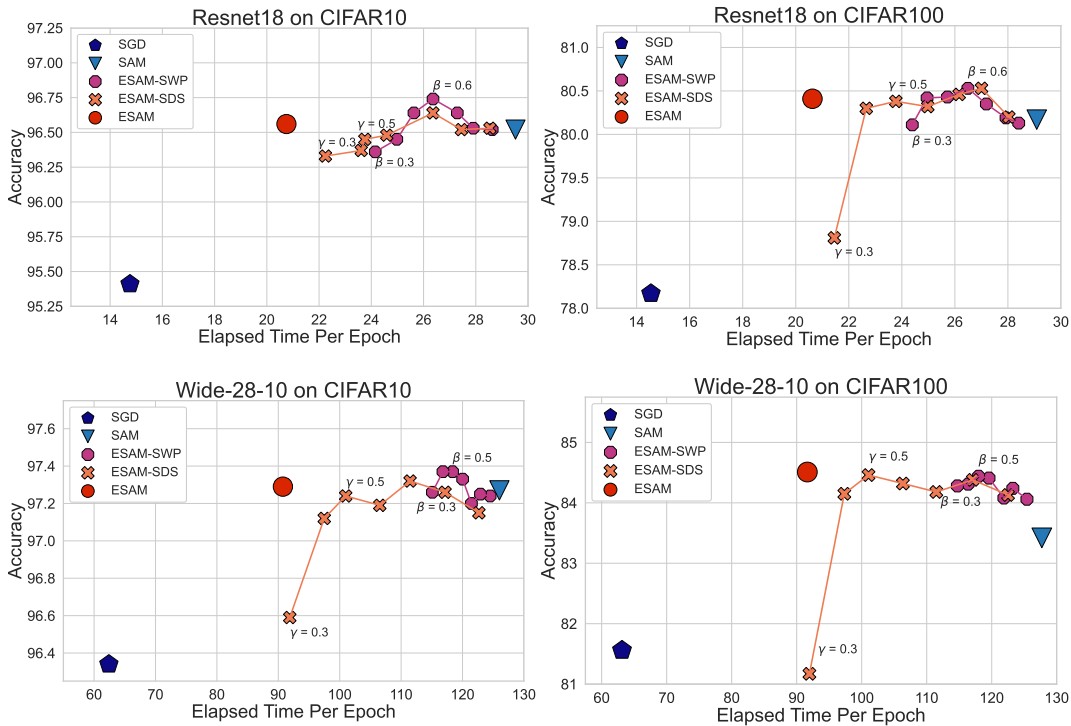

Figure 3: Parameter study of SWP and SDS. The connected dots refer to SWP and SDS with different parameters; the isolated dots refer to the final results of SGD, SAM, and ESAM.

**ESAM-SWP** As shown in Table 3, SWP improves SAM's training speed by 8.3% to 10.5%, and achieves better performance at the same time. SWP can further improve the efficiency by using a smaller $\beta$. The best performance of SWP is obtained when $\beta = 0.6$ for ResNet-18 and $\beta = 0.5$ for WideResNet-28-10. The four sets of experiments indicate that $\beta$ is consistent among different architectures and datasets. Therefore, we set $\beta = 0.6$ for PyramidNet on CIFAR10/100 datasets and ResNet on ImageNet datasets.

**ESAM-SDS** SDS also significantly improves the efficiency by 21.5% to 25.8% compared to SAM. It outperforms SAM's performance on CIFAR100 datasets, and achieves comparable performance on CIFAR10 datasets. SDS can outperform SAM on both datasets with both architectures with little degradation to the efficiency, as demonstrated in Figure 3. Across all experiments, $\gamma = 0.5$ is the smallest value that is optimal for efficiency while maintaining comparable performance to SAM.

**Visualization of Loss Landscapes** To visualize the sharpness of the flat minima obtained by ESAM, we plot the loss landscapes trained with SGD, SAM and ESAM on the ImageNet dataset. We display the loss landscapes in Figure 4, following the plotting algorithm in Li et al. (2018). The $x$- and $y$-axes represent two random sampled orthogonal Gaussian perturbations. We sampled $100 \times 100$ points for 10 groups random Gaussian perturbations. The displayed loss landscapes are the results we obtained by averaging over ten groups of random perturbations. It can be clearly seen that both SAM and ESAM improve the sharpness significantly in comparison to SGD.

To summarize, SWP and SDS both reduce the computational overhead and accelerate training compared to SAM. Most importantly, both these strategies achieve a comparable or better performance than SAM. In practice, by configuring the $\beta$ and $\gamma$, ESAM can meet a variety of user-defined efficiency and performance requirements.

## 4 RELATED WORK

The concept of regularizing sharpness for better generalization dates back to (Hochreiter & Schmidhuber, 1995). By using an MDL-based argument, which clarifies that a statistical model with fewer

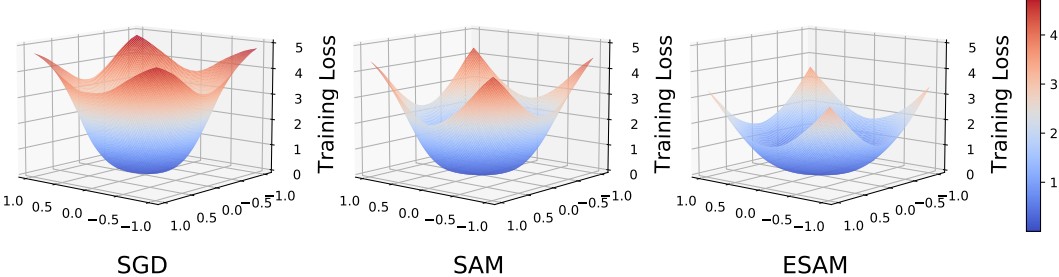

Figure 4: Cross-entropy loss landscapes of the ResNet50 model on the ImageNet dataset trained with SGD, SAM, and ESAM.

bits to describe can have better generalization ability, Hochreiter & Schmidhuber (1995) claim that a flat minimum can alleviate overfitting issues. Following that, more studies were proposed to investigate the connection between the flat minima with the generalization abilities (Keskar et al., 2017; Dinh et al., 2017; Liu et al., 2020; Li et al., 2018; Dziugaite & Roy, 2017; Jiang et al., 2019; Moosavi-Dezfooli et al., 2019). Keskar et al. (2017) starts by investigating the phenomenon that training with a larger batch size results in worse generalization ability. The authors found that the sharpness of the minimum is critical in accounting for the observed phenomenon. Keskar et al. (2017) and Dinh et al. (2017) both argue that the sharpness can be characterized using the eigenvalues of the Hessian. Although they also define specific notions and methods to quantify sharpness, they do not propose complete training strategies to find minima that are relative "flat".

SAM (Foret et al., 2020) leverages the connection between "flat" minima and the generalization error to train DNNs that generalize well across the natural distribution. Inspired by Keskar et al. (2017) and Dinh et al. (2017), SAM first proposes the quantification of the sharpness, which is achieved by solving a maximization problem. Then, SAM proposes a complete training algorithm to improve the generalization abilities of DNNs. SAM is demonstrated to achieve state-of-the-art performance in a variety of deep learning benchmarks, including image classification, natural language processing, and noisy learning (Foret et al., 2020; Chen et al., 2021; Kwon et al., 2021; Pham et al., 2021; Yuan et al., 2021; Jia et al., 2021).

A series of SAM-related works has been proposed. A work that was done contemporaneously SAM (Wu et al., 2020) also regularizes the sharpness term in adversarial training and achieves much more robust generalization performance against adversarial attacks. Many works focus on combining SAM with other training strategies or architectures (Chen et al., 2021; Wang et al., 2022; Tseng et al., 2021), or apply SAM on other tasks (Zheng et al., 2021; Damian et al., 2021; Galatolo et al., 2021). Kwon et al. (2021) improves SAM's sharpness by adaptively scaling the size of the nearby search space $\rho$ in relation to the size of parameters. Liu et al. (2022) leverages the past calculated weight perturbations to save SAM's computations. However, most of these works overlook the fact that SAM improves generalization at the expense of the doubling the computational overhead. As a result, most of the SAM-related works suffer from the same efficiency drawback as SAM. This computational cost prevents SAM from being widely used in large-scale datasets and architectures, particularly in real-world applications, which motivates us to propose ESAM to *efficiently* improve the generalization ability of DNNs.

## 5 CONCLUSION

In this paper, we propose the *Efficient Sharpness Aware Minimizer* (ESAM) to enhance the efficiency of vanilla SAM. The proposed ESAM integrates two novel training strategies, namely, SWP and SDS, both of which are derived based on theoretical underpinnings and are evaluated over a variety of datasets and DNN architectures. Both SAM and ESAM are two-step training strategies consisting of sharpness estimation and weight updating. In each step, gradient back-propagation is performed to compute the weight perturbation or updating. In future research, we will explore how to combine the two steps into one by utilizing the information of gradients in previous iterations so that the computational overhead of ESAM can be reduced to the same as base optimizers.

ACKNOWLEDGEMENT

Jiawei Du and Joey Tianyi Zhou are supported by Joey Tianyi Zhou's A*STAR SERC Central Research Fund.

Hanshu Yan and Vincent Tan are funded by a Singapore National Research Foundation (NRF) Fellowship (R-263-000-D02-281) and a Singapore Ministry of Education AcRF Tier 1 grant (R-263-000-E80-114).

We would like to express our special thanks of gratitude to Dr. Yuan Li for helping us conduct experiments on ImageNet, and Dr. Wang Yangzihao for helping us implement Distributed Data Parallel codes.

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

## A APPENDIX

### A.1 THE ALGORITHM OF SAM

The algorithm of SAM is demonstrated in Algorithm 2.

---

**Algorithm 2** SGD vs. SAM

---

**Input:** Network $f_\theta$, $\theta = (\theta_1, \theta_1, \ldots, \theta_N)$, Training set $\mathbb{S}$, Batch size $b$, Learning rate $\eta$, Neighborhood size $\rho$, Iterations $A$.
**Output:** A minimum solution $\tilde{\theta}$.
 1: **for** $a = 1$ to $A$ **do**
 2:      Sample $\mathbb{B}$ with size b that $\mathbb{B} \subset \mathbb{S}$,
 3:      **if** SGD **then**
 4:          $\hat{\epsilon} \leftarrow 0$                      ▷ No addtional computational overhead for $\hat{\epsilon}$
 5:      **else if** SAM **then**
 6:          $\hat{\epsilon} \leftarrow \nabla_\theta L_\mathbb{B}(f_\theta)$              ▷ additional $F_1$ and $B_1$ to compute $\hat{\epsilon}$
 7:      Compute $g = \nabla_\theta L_\mathbb{B}(f_{\theta+\hat{\epsilon}})$                        ▷ $F_2$ and $B_2$
 8:      update the weights $\theta \leftarrow \theta - \eta g$

---

### A.2 OPTIMIZING OVER SUBSET $\mathbb{B}^+$ IS REPRESENTATIVE

The sharpness-sensitive subset $\mathbb{B}^+$ is constructed by sorting $\ell(f_{\theta+\hat{\epsilon}}, x_i, y_i) - \ell(f_\theta, x_i, y_i)$, which is positively correlated to $l(f_\theta, x_i, y_i)$. By a first-order Taylor series approximation,

$$\ell(f_{\theta+\hat{\epsilon}}, x_i, y_i) - \ell(f_\theta, x_i, y_i) = \hat{\epsilon} \cdot \nabla_\theta \ell(f_\theta, x_i, y_i) + o(\|\hat{\epsilon}\|).$$

By Equation 4, $\hat{\epsilon}$ is the aggregated gradients of each instance in the complete dataset $\mathbb{B}$, i.e.,

$$\hat{\epsilon} = \arg\max_{\epsilon: \|\epsilon\|_2 < \rho} L_\mathbb{S}(f_{\theta+\epsilon}) \approx \rho \nabla_\theta L_\mathbb{S}(f_\theta) = \sum_{i=1}^{|\mathbb{B}|} \nabla_\theta \ell(f_\theta, x_i, y_i),$$

which indicates that $\ell(f_{\theta+\hat{\epsilon}}, x_i, y_i) - \ell(f_\theta, x_i, y_i)$ is positively correlated to the gradient $\nabla_\theta \ell(f_\theta, x_i, y_i)$. Li et al. (2019) claims that the difficult examples in deep learning (the training samples with high training loss) produce gradients with larger magnitudes. Therefore, $\ell(f_{\theta+\hat{\epsilon}}, x_i, y_i) - \ell(f_\theta, x_i, y_i)$ is positively correlated to $l(f_\theta, x_i, y_i)$. We also demonstrate the correlation empirically.

We conduct experiments to verify Equation 9 and Equation 10. In Figure 5, We plot the four losses, $L_{\mathbb{B}^+}(f_\theta)$, $L_{\mathbb{B}^-}(f_\theta)$, $L_{\mathbb{B}^+}(f_{\theta+\hat{\epsilon}})$, and $L_{\mathbb{B}^-}(f_{\theta+\hat{\epsilon}})$ w.r.t the epochs. The experimental results verify that Equation 9 and Equation 10 hold for every training epoch.

Moreover, we conduct experiments to demonstrate that optimizing over the subset $\mathbb{B}^+$ is much more representative than the subset $\mathbb{B}^-$. We compare the updating gradients computed from $\mathbb{B}^+, \mathbb{B}^-$ and a random subset $\mathbb{B}_{rand}$ that $|\mathbb{B}_{rand}| = |\mathbb{B}^+| = |\mathbb{B}^-|$ to those computed from $\mathbb{B}$ by calculating the cosine similarity inspired by (Du et al., 2019), i.e.

$$\text{CosSim}(\nabla_\theta L_{\mathbb{B}^+}(f_{\theta+\hat{\epsilon}}), \nabla_\theta L_\mathbb{B}(f_{\theta+\hat{\epsilon}})),$$

$$\text{CosSim}(\nabla_\theta L_{\mathbb{B}^-}(f_{\theta+\hat{\epsilon}}), \nabla_\theta L_\mathbb{B}(f_{\theta+\hat{\epsilon}})),$$

$$\text{CosSim}(\nabla_\theta L_{\mathbb{B}_{rand}}(f_{\theta+\hat{\epsilon}}), \nabla_\theta L_\mathbb{B}(f_{\theta+\hat{\epsilon}})).$$

In Figure 6, we plot the cosine similarities in each training epoch evaluated with ResNet-18, Wide-28-10 on CIFAR10. In terms of the computed gradients, the experimental results show that $\mathbb{B}^+$ has the highest cosine similarities with $\mathbb{B}$ than $\mathbb{B}^-$ and the random set $\mathbb{B}_{rand}$.

### A.3 LINEARITY MEASUREMENT OF SWP

As the experimental results in Figure 3 demonstrated, SWP can also improve the accuracy of ESAM compared to SAM. We will investigate the advantage of SWP in terms of generalization in the

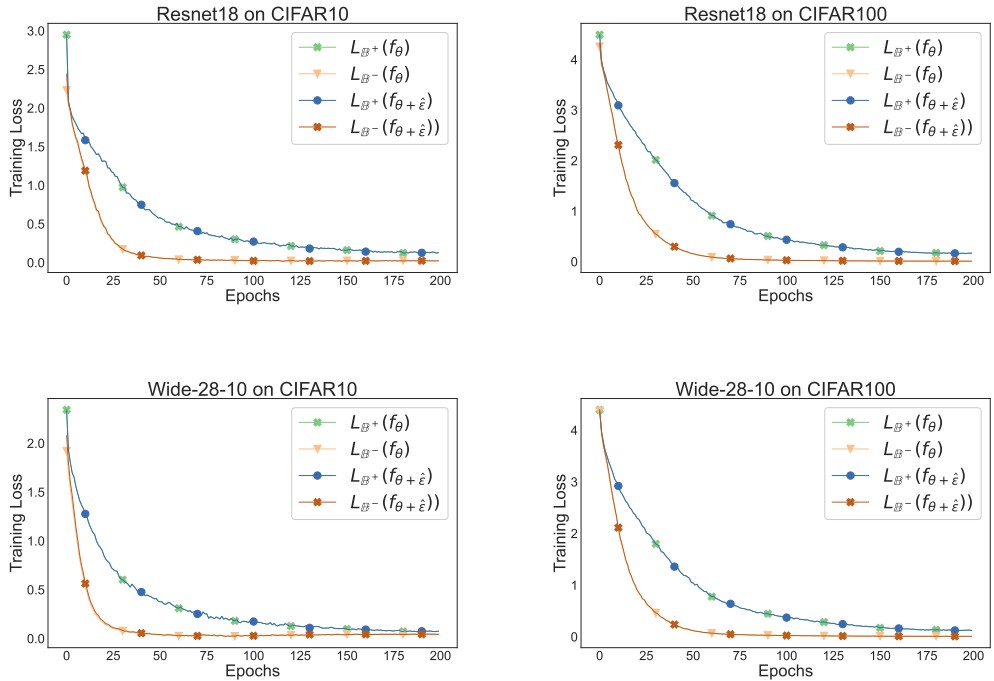

Figure 5: The SAM loss and the empirical loss calculated over the selected subsets $\mathbb{B}^+$, $\mathbb{B}^-$, w.r.t the epochs, as evaluated with ResNet-18, Wide-28-10 on CIFAR10 and CIFAR100. The subset $\mathbb{B}^+$ selected by SDS has much higher SAM loss and empirical loss than $\mathbb{B}^-$ among all the four groups of experiments.

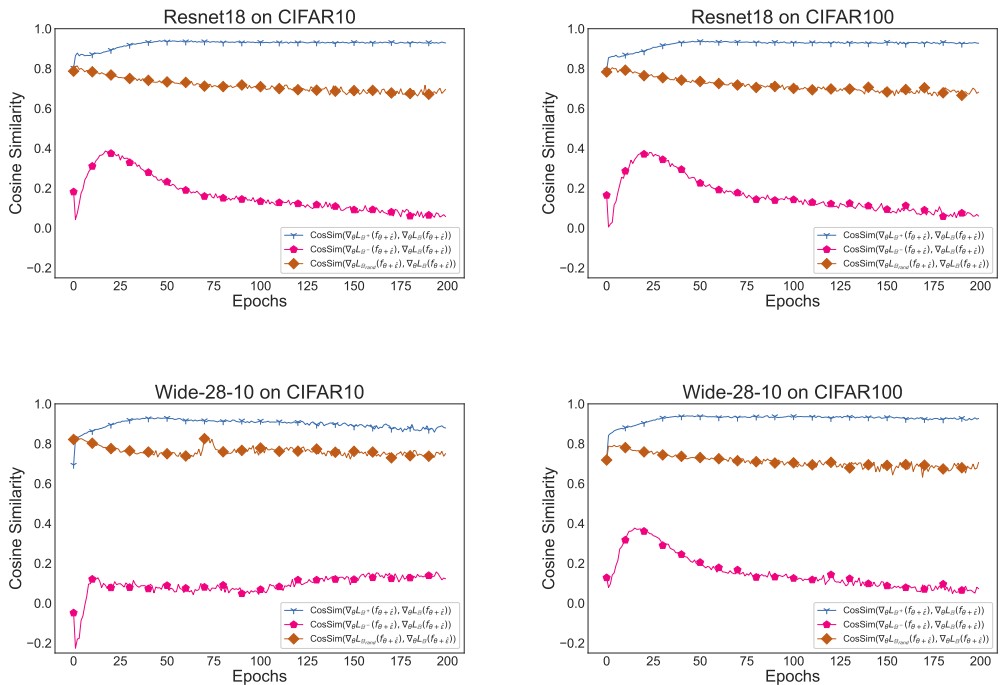

Figure 6: The cosine similarity between the gradients computed from subsets $\mathbb{B}^+$, $\mathbb{B}^-$, and $\mathbb{B}_{\text{rand}}$ with $\mathbb{B}$, as evaluated with ResNet-18, Wide-28-10 on CIFAR10 and CIFAR100. The gradients from the subset $\mathbb{B}^+$ selected by SDS has much higher cosine similarity with the gradients from $\mathbb{B}$ than the gradients from $\mathbb{B}^-$ and $\mathbb{B}_{\text{rand}}$ among all the four groups of experiments.

future research. Here we provide a discussion about the accuracy improvement contributed by SWP. A plausible reason for such improvement is that SWP leads to a better inner maximization solved in equation 2. The current solution of $\hat{\epsilon}$ is approximated by assuming $L_\mathbb{S}(f_\theta)$ is a linear function. Therefore, the $\hat{\epsilon}$ would result in a better inner maximization if $L_\mathbb{S}(f_\theta)$ is "more linear" with respect to $\theta$. Inspired by (Qin et al., 2019; Yan et al., 2019; 2021), we measure the linearity of the loss function by

$$\zeta(\epsilon, \mathbb{B}) = |L_\mathbb{B}(f_{\theta+\epsilon}) - L_\mathbb{B}(f_\theta) - \epsilon^\top \nabla_\theta L_\mathbb{B}(f_\theta)|.$$

We conduct experiments on the CIFAR10 dataset with ResNet-18 model to verify that SWP can improve the linearity $\zeta(\epsilon, \mathbb{B})$ of the loss function $L_\mathbb{S}(f_\theta)$. We compare the linearity of ESAM with the $\beta$ ranging from $\{0.2, 0.3, ..., 0.9\}$ to the SAM. The results are demonstrated in Figure 7. It can

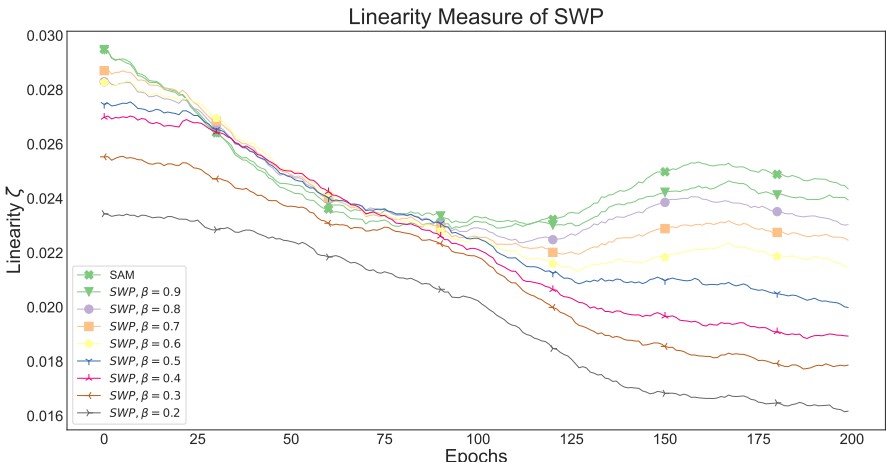

Figure 7: The linearity measurement of SWP evaluated with the ResNet18 model on the CIFAR10 dataset compared with SAM. The experimental results indicate that a smaller $\beta$ can result in better linearity.

be shown that SWP will result in a better linearity as the $\beta$ decreases. However, decreasing $\beta$ will also reduce the magnitude of $\hat{\epsilon}$ and thus result in a worse inner maximization in equation 2. We observe that $\beta = \{0.5, 0.6\}$ is optimal to balance the accuracy and efficiency of ESAM.

### A.4 REDUCED COMPUTATIONAL OVERHEAD CONTRIBUTED BY SWP

**Formulation** We formulate the saved computational overhead contributed by SWP here. We discuss and examine the computational overhead in the PyTorch framework. Suppose that the NN we discussed here has $N$ layers. The most unit of parameters ($D \times C \times H \times W$) is the entire parameters of a layer in NN, where $D$ is the number of kernels, $C$ is the number of channels, $H$ and $W$ are the height and width of the input. SWP select each basic parameter unit to compute gradients (i.e. requries_grad = True) with probability $\beta$. We use $g(N, \beta)$ to measure the saved computational overhead in terms of percentage contributed by SWP compared to the vanilla SAM.

The computational overhead $g(N, \beta)$ is reduced not just by calculating gradients, but also by the storing and hooking gradients. Because the storing and hooking operations of each parameter's gradients are independent to each other, the computational overhead saved from storing and hooking are proportional to $1 - \beta$ and irrelevant to the depth $N$ of NN. In addition, the storing and hooking gradients are the dominant factor that result in the reduced computational overhead according to our toy example in the following.

Next we discuss the saved computational overhead that stems from the calculation operations in the general case. Some layers in the DNNs with complicated architectures such as DenseNet and ViT, may have multiple basic parameter units in the same layer and be connected across any other layers. Suppose the $n_{th}$ layer has $K(n)$ basic parameter units, let $p^n$ be the calculation-free rate of a certain parameter unit in $n_{th}$ layer, we have $p^n = (1-\beta)^{K(n)} \cdot p^{n-1}$. Therefore, $p^n = (1-\beta)^{\sum_{j=1}^n K(j)} \approx$

$(1 - \beta)^{n\bar{K}}$, where $\bar{K} = \frac{1}{N} \sum_{j=1}^{N} K(j)$. Assumed that the computations of each layer are the same, by summing up the saved computational overhead of all parameters, we have

$$
\begin{aligned}
g(N, \beta) &= k_1(1 - \beta) + k_2 \sum_{n=i}^{N} \frac{p^n}{N} \\
&\approx k_1(1 - \beta) + \frac{k_2(1 - \beta)}{N[1 - (1 - \beta)^{\bar{K}}]} \\
&\approx k_1(1 - \beta) + \frac{k_2'(1 - \beta)}{N\beta}.
\end{aligned}
\tag{11}
$$

where $k_1$, $k_2$ are determined by the computing time of calculation, $k_2' = \frac{k_2 \beta}{[1 - (1 - \beta)^{\bar{K}}]}$.

However, the commonly used DNNs' architectures such as ResNet only have one basic parameter unit in each layer. Besides, each layer of them is connected in serial with the next layer. Then, we have $p^n = (1 - \beta) \cdot p^{n-1}$. Therefore, $p^n = (1 - \beta)^n$. The saved calculation contributed by the parameter unit is $\frac{1}{N}p^n$. By summing up the saved computational overhead of all parameters, we have

$$
\begin{aligned}
g(N, \beta) &= k_1(1 - \beta) + k_2 \sum_{n=i}^{N} \frac{p^n}{N} \\
&= k_1(1 - \beta) + \frac{k_2}{N} \frac{1 - \beta - (1 - \beta)^N}{\beta} \\
&\approx k_1(1 - \beta) + \frac{k_2(1 - \beta)}{N\beta}.
\end{aligned}
\tag{12}
$$

where $k_1$, $k_2$ are determined by the computing time of calculation, storing and hooking gradients.

**Toy Example** We conducted a toy example on the CIFAR10 dataset with two MLPs. Each fully connected layer of MLP is the same with a size of $3,000$ for both in and out features. The first mlp is for the special case that $\bar{K} = 1$, where each layer has only one basic parameter unit and is connected in serial with the next layer. We examined $N = \{50, 75, 100, 125\}$ and $\beta = \{0.1, ..., 0.9, 1.0\}$ to record the saved computational overhead in percentage. Part of the results are reported in Table A.4. By linear regression, we have $k_1 = 0.3185, k_2 = 0.1310$, and the returned $R^2 = 0.9983$. The second mlp is for the general case that $\bar{K} > 1$, where each layer may have multiple basic parameter units in the same layer and be connected across any other layers. We examined $N = \{35, 50, 65, 75\}$ and $\beta = \{0.1, ..., 0.9, 1.0\}$ to record the saved computational overhead in percentage. Part of the results are reported in Table A.4. By linear regression, we have $k_1 = 0.3143, k_2 = 0.0737$, and the returned $R^2 = 0.9989$. The above experimental results verify the formulation of the reduced computational overhead contributed by SWP in equation 11 and equation 12.

| $N$ | $\beta$ | $g(N, \beta)$ |
|-----|---------|---------------|
| 50  | 0.9     | 3.42%         |
| 50  | 0.5     | 16.28%        |
| 50  | 0.1     | 30.08%        |
| 75  | 0.9     | 3.44%         |
| 75  | 0.5     | 15.79%        |
| 75  | 0.1     | 30.42%        |
| 100 | 0.9     | 3.37%         |
| 100 | 0.5     | 15.86%        |
| 100 | 0.1     | 29.56%        |
| 125 | 0.9     | 2.94%         |
| 125 | 0.5     | 14.87%        |
| 125 | 0.1     | 29.42%        |

| $N$ | $\beta$ | $g(N, \beta)$ |
|-----|---------|---------------|
| 25  | 0.9     | 2.88%         |
| 25  | 0.5     | 15.54%        |
| 25  | 0.1     | 30.03%        |
| 50  | 0.9     | 2.34%         |
| 50  | 0.5     | 15.10%        |
| 50  | 0.1     | 29.83%        |
| 65  | 0.9     | 2.23%         |
| 65  | 0.5     | 14.95%        |
| 65  | 0.1     | 29.22%        |
| 75  | 0.9     | 2.22%         |
| 75  | 0.5     | 15.10%        |
| 75  | 0.1     | 28.68%        |

Table 4: The special case that $\bar{K} = 1$. By linear regression, $k_1 = 0.3185, k_2 = 0.1310$, and the returned $R^2 = 0.9983$.

Table 5: The general case that $\bar{K} > 1$. By linear regression, $k_1 = 0.3143, k_2' = 0.0737$, and the returned $R^2 = 0.9989$.

## A.5 VISUALIZATION OF LOSS LANDSCAPES WITH RESPECT TO ADVERSARIAL WEIGHT PERTURBATIONS

We visualize the sharpness of the flat minima with respect to adversarial weight perturbations of SGD,SAM and ESAM on the Cifar10 dataset. The $x$- and $y$-axes represent two orthogonal adversarial weight perturbations, which are $\eta\nabla_\theta L_{\mathbb{B}_x}(f_\theta)$ and $\eta\nabla_\theta L_{\mathbb{B}_y}(f_\theta)$ respectively, where $\eta$ is the learning rate during training. $\mathbb{B}_x$ and $\mathbb{B}_y$ are the randomly sampled subsets of batch $\mathbb{B}$, and $|\mathbb{B}_x| = |\mathbb{B}_y| = \frac{1}{2}|\mathbb{B}|$, $\mathbb{B}_x \cup \mathbb{B}_y = \mathbb{B}$. We display the loss landscape in Figure 8, which demonstrates that both SAM and ESAM improve the sharpness significantly in comparison to SGD.

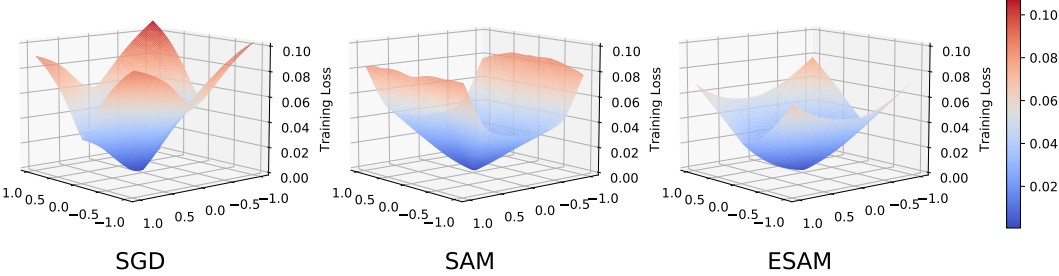

Figure 8: Cross-entropy loss landscapes of the ResNet18 model respect to adversarial weight perturbations on the CIFAR10 dataset trained with SGD, SAM, and ESAM.

## A.6 EVALUATION OF ESAM ON VIT-S/16

SAM has also been demonstrated to be effective on the new vision Transformer(ViT) architecture (Chen et al., 2021). Therefore, we also evaluate ESAM with ViT-S/16 on ImageNet Datasets. We use $\beta = 0.5$ and $\gamma = 0.7$ for ViT-S/16, which share the same hyperparameters as ResNet-50 and ResNet-101 in section 3.1. The results are reported in Table 6, which indicate that ESAM can still be effective to improve efficiency in ViT-S/16 architecture. In particular, ESAM-SWP achieves much better accuracy than SAM (80.88% v.s. 80.34%).

Table 6: Classification accuracies and training speed of ViT-S/16 on the ImageNet dataset.

| | ViT-S/16 | |
|---|---|---|
| ImageNet | Accuracy | images/s |
| SGD | 79.72 | 1,133 |
| SAM | 80.34 | 581 |
| ESAM-SWP | 80.88 | 616 |
| ESAM-SDS | 79.97 | 693 |
| ESAM | 80.46 | 734 |

## A.7 TRAINING DETAILS

We tune the training parameters of SGD, SAM, and ESAM, by using grid searches. The learning rate is chosen from the set $\{0.01, 0.05, 0.1, 0.2\}$, the weight decay from the set $\{5 \times 10^{-4}, 1 \times 10^{-3}\}$, and the batch size from the set $\{64, 128, 256\}$. This is done to attain the best accuracies. The exact training hyperparameters are reported in Table 7. On the ImageNet datasets, limited by the computing resource, we follow and slightly modify the optimal hyperparameters as suggested by Chen et al. (2021) for SGD, SAM and ESAM. The exact training hyperparameters are reported in Table 8.

Table 7: Hyperparameters for training from scratch on CIFAR10 and CIFAR100

| | CIFAR-10 | | | CIFAR-100 | | |
|---|---|---|---|---|---|---|
| **ResNet-18** | SGD | SAM | ESAM | SGD | SAM | ESAM |
| Epoch | | 200 | | | 200 | |
| Batch size | | 128 | | | 128 | |
| Data augmentation | | Basic | | | Basic | |
| Peak learning rate | | 0.05 | | | 0.05 | |
| Learning rate decay | | Cosine | | | Cosine | |
| Weight decay | $5 \times 10^{-4}$ | $1 \times 10^{-3}$ | $1 \times 10^{-3}$ | $5 \times 10^{-4}$ | $1 \times 10^{-3}$ | $1 \times 10^{-3}$ |
| $\rho$ | - | 0.05 | 0.05 | - | 0.05 | 0.05 |
| **Wide-28-10** | SGD | SAM | ESAM | SGD | SAM | ESAM |
| Epoch | | 200 | | | 200 | |
| Batch size | | 256 | | | 256 | |
| Data augmentation | | Basic | | | Basic | |
| Peak learning rate | | 0.05 | | | 0.05 | |
| Learning rate decay | | Cosine | | | Cosine | |
| Weight decay | $5 \times 10^{-4}$ | $1 \times 10^{-3}$ | $1 \times 10^{-3}$ | $5 \times 10^{-4}$ | $1 \times 10^{-3}$ | $1 \times 10^{-3}$ |
| $\rho$ | - | 0.1 | 0.1 | - | 0.1 | 0.1 |
| **PyramidNet-110** | SGD | SAM | ESAM | SGD | SAM | ESAM |
| Epoch | | 300 | | | 300 | |
| Batch size | | 256 | | | 256 | |
| Data augmentation | | Basic | | | Basic | |
| Peak learning rate | | 0.1 | | | 0.1 | |
| Learning rate decay | | Cosine | | | Cosine | |
| Weight decay | | $5 \times 10^{-4}$ | | | $5 \times 10^{-4}$ | |
| $\rho$ | - | 0.2 | 0.2 | - | 0.2 | 0.2 |

Table 8: Hyperparameters for training from scratch on ImageNet

| | ResNet-50 | | | ResNet-110 | | |
|---|---|---|---|---|---|---|
| **ImageNet** | SGD | SAM | ESAM | SGD | SAM | ESAM |
| Epoch | | 90 | | | 90 | |
| Batch size | | 512 | | | 512 | |
| Data augmentation | | Inception-style | | | Inception-style | |
| Peak learning rate | | 0.2 | | | 0.2 | |
| Learning rate decay | | Cosine | | | Cosine | |
| Weight decay | | $1 \times 10^{-4}$ | | | $1 \times 10^{-4}$ | |
| $\rho$ | - | 0.05 | 0.05 | - | 0.05 | 0.05 |
| Input resolution | | $224 \times 224$ | | | $224 \times 224$ | |

