# OpenReview forum: "Efficient Sharpness-aware Minimization for Improved Training of Neural Networks"
_ICLR.cc/2022/Conference — ICLR 2022 Poster_

### Official Review · Reviewer_cAHr · 2021-10-31

**Correctness:** 4
**Technical Novelty And Significance:** 4
**Empirical Novelty And Significance:** 4
**Recommendation:** 8
**Confidence:** 4

**Main Review:**

Strengths:
* This paper is well written and easy to follow.
* While SAM has proven to be helpful for many applications, this technique can be helpful in practice.

Weakness:
*  For SWP, is the gradient mask the same for the samples in one batch, or different gradient masks are applied for each sample in the batch?
* For SDS, an interesting baseline to be compared is estimating $\epsilon$ with a subset of randomly sampled data, which could match the origin $\epsilon$ in terms of empirical estimation.
*  Recent paper [1] points that SAM can be very useful in ViT and MLP-Mixer. It would be more convincing if the author could show ESAM can also work on those two architectures.

Detailed comments:
* First row in Page 8,  "As shwon in Table 3, SWP improves". shwon -> shown

[1] Chen, Xiangning, Cho-Jui Hsieh, and Boqing Gong. "When Vision Transformers Outperform ResNets without Pretraining or Strong Data Augmentations." arXiv preprint arXiv:2106.01548 (2021).

**Summary Of The Paper:**

This paper presents a method called ESAM for improving the efficiency of SAM. ESAM has two components: SWP and SDS. SWS accelerates the estimation of $\epsilon$ via random sampling a subset of parameters in backpropagation. SDS further improves the efficiency via sampling a subset of data points that is enough for calculating the upper bound of $L$. Combining these two techniques, they achieve a speed improvement for SAM while yielding comparable or even better performance.

**Summary Of The Review:**

This paper is the pioneer in exploring a more efficient SAM method. This technique can be useful since SAM has been proven to be useful for many networks. Therefore, I would recommend this paper as weak accept. I would be more convincing the practicability of this paper if the authors can show the proposed ESAM works on ViT and MLP mixer, where the SAM has been proven to be very effective.

---

> ### Author Response · Authors · 2021-11-20
> **Response**
>
> We appreciate your valuable comments, and answer your questions in order.
>
> **Question 1:** For SWP, is the gradient mask the same for the samples in one batch, or different gradient masks are applied for each sample in the batch?
>
> **Answer:** The gradient mask is the same for the samples in one batch. However, the gradient mask will not be the same among different GPUs in Distributed Data Parallel (DDP) training. We implement the SWP by randomly setting the "requires_grads" of kernels' parameters to false in each layer.  A kernel's parameters tensor is the basic unit that can be operated in the PyTorch framework. We resample these kernels randomly in each iteration. The implementation codes of SWP have been attached in the supplementary files.
>
>
> **Question 2:** An interesting baseline to be compared is estimating ϵ with a subset of randomly sampled data
> **Answer:** Thank you for the constructive suggestions; we added the random set baseline that replaces the selected $\mathbb{B}^+$ by a randomly sampled subset $\mathbb{B}_{\text{rand}}$ and reported the following results.
>
> |ResNet-18 | CIFAR-10 | CIFAR-100|
> | ----------- | :-----------: |:-----------:|
> |SGD | 95.63 | 78.69 |
> |SAM | 96.51 | 80.17|
> |SDS random set | 96.03 | 79.97|
> |ESAM-SDS(ours)| 96.45 | 80.38|
> |ESAM(ours) | 96.56 | 80.41|
>
>
> |Wide-28-10 |     CIFAR-10 | CIFAR-100|
> | ----------- | :-----------: |:-----------:|
> |SGD|96.56|81.98|
> |SAM|97.27|83.42|
> |SDS random set |96.89 | 83.01|
> |ESAM-SDS(ours)| 97.24 | 84.46 |
> |ESAM(ours)|97.29|84.51|
>
>
> As the results demonstrated, replacing the selected $\mathbb{B}^+$ by a randomly sampled subset $\mathbb{B}_{\text{rand}}$ will lead to a serious accuracy degradation on both CIFAR-10 and CIFAR-100 datasets.
>
> Moreover, we also measure the gradients' alignment computed by $\mathbb{B}^+$, $\mathbb{B}^-$ and $\mathbb{B}\_{\text{rand}}$ with the gradients computed by the full-batch $\mathbb{B}$ in terms of cosine similarity. We report the results in appendix A.2 Figure 5. It can be shown that the gradients computed by $\mathbb{B}^+$ have the highest cosine similarities with the full-batch $\mathbb{B}$ than $\mathbb{B}^-$ and $\mathbb{B}\_{\text{rand}}$.
>
> **Question:** It would be more convincing if the author could show ESAM can also work on those two architectures.
>
> **Answer:** Thank you for your suggestions. We conduct experiments with VIT-S/16 on ImageNet datasets. We use $\beta=0.5$ and $\gamma=0.7$ for VIT-S/16, which is the same with the settings of ResNets on ImageNet datasets. We report the results as follows,
>
> |ViT-S/16 | accuracy | Images/s |
> | ----------- | :-----------: |:-----------:|
> |SGD | 79.72 | 1,133 |
> |SAM | 80.34 | 581 |
> |ESAM-SWP | 80.88 | 616 |
> |ESAM-SDS | 79.97 | 693 |
> |ESAM | 80.46 | 734 |
>
> The experimental results demonstrate that ESAM can still be effective to improve efficiency in ViT-S/16 architecture. In particular, ESAM-SWP achieves much better accuracy than SAM(80.88% v.s. 80.34%). We did not have time to conduct experiments on the MLP mixer and will add the results once finished.  We append the settings of ViT-S/16 as follows for reproductivity.
>
> “CUDA_VISIBLE_DEVICES=0,1,2,3 ./distributed_train.sh 4 --model vit_small_patch16_224 -b 256 --lr 1e-3 --sched cosine --epochs 300 --opt adamw -j 8 --warmup-lr 1e-6 --mixup .8 --aa rand-m9-mstd0.5-inc1 --remode pixel --reprob 0.25 --cutmix 1.0 --weight-decay .05 --drop 0.0 --drop-path .1 --warmup-epochs 5 --img-size 224 --beta 0.5 --gamma 0.7  \rho 0.1”

---

> > ### Comment · Reviewer_cAHr · 2021-11-23
> > **My concern has been addressed**
> >
> > The authors' excellent rebuttal addresses all my concerns. I have raised my score from 6->8.

---

### Official Review · Reviewer_wp3y · 2021-11-02

**Correctness:** 4
**Technical Novelty And Significance:** 3
**Empirical Novelty And Significance:** 3
**Recommendation:** 6
**Confidence:** 4

**Main Review:**

Pros:
1.	The proposed approach is well-motivated and practical in real-world applications. The enhanced efficiency does not lead to a degradation of accuracy. The experimental setup is satisfactory and convincing. The proposed method is verified on multiple datasets to show the effectiveness. The ablation studies and the supplementary experiments in the appendix are appropriate.
2.	The idea of picking representative samples for updating parameters is interesting and effective. Why not use the $\alpha$ directly as the hyperparameter for data selection?
3.	The paper is well-organized. Figures 3 and 4 are good for demonstrating.
Cons:
1.	It is mentioned that the computations of SWP slightly increase in the deeper DNNs. What is the exact impact of using SWP in large-scale deep neural networks?
2.	The explanation of SWP's computation in Section 2.2 is not clear and may be made shorter. For example, L8 "Decreasing $\beta$ ... may degrade" is superfluous; The claimed positive correlation between saved computations with $\beta$ is vague and should be more precise.
3.	The gaussian perturbations used in Figure 4 are not representative enough. There is no difference between the x-axis perturbation and the y-axis perturbation. Consider visualizing the loss landscape with the adversarial perturbations.
Additional comments:
1.	Better to use "hyperparameters of SGD" to replace "parameters of SGD" in sec 3.1 line 6.
2.	Better to add "(Training speed 140.3% vs. SAM 100%)" for the efficiency comparison in section 3.1 P3L4 "(140.3% vs. SAM 100%)".


**Summary Of The Paper:**

This paper investigates the crucial efficiency issue of the sharpness-aware minimizer (SAM). SAM improves the generalization of DNNs but results in a double training time compared to vanilla training. By analyzing the min-max procedure of SAM, the authors observe the computational redundancy and then propose a method ESAM to improve the efficiency from the data and parameters perspectives. The authors argue that SAM can be approximated properly with fewer computations. Empirical results show that ESAM can reduce the extra training time in CIFAR10/100 and ImageNet datasets with improved accuracy compared to SAM.

**Summary Of The Review:**

This work raises an important question of SAM where the improved training algorithm requires doubled training cost. This paper addresses and improves the efficiency drawback of SAM. The proposed approach is well-motivated and has been verified to improve the efficiency and accuracy of SAM. The authors also provide codes for reproducibility. However, the degradation of SWP's effectiveness in large-scale neural networks, as well as the justifications for the improved accuracy contributed by SWP and SDS, have not been clearly addressed in this paper. I expect the author can answer the points above, and then I can adjust my final score.

---

> ### Author Response · Authors · 2021-11-20
> **Response**
>
> Thank you for your thoughtful review. We answer your questions in the following.
>
> **Question 1:** Why not use $\alpha$ directly as the hyperparameter for data selection?
>
> **Answer:**
> Because  the distributions of instance-wise loss differences in equation 6 vary on different model architectures and datasets, it is hard to tune $alpha$ to a proper value. Therefore, we use $\gamma = \frac{|\mathbb{B}^+|}{|\mathbb{B}|}$ as the hyperparameter of SDS for better tuning.
>
> **Question2 :** What is the exact impact of using SWP in large-scale deep neural networks?
>
> **Answer:** Thanks a lot for the constructive comments. We formulate the impact of SWP on efficiency in Appendix A.4. To summarize, the reduced training time contributed by SWP is approximately linear with $\beta$ and $\frac{1-\beta}{N\beta}$, where N is the depth of neural networks. We also presented a prototype example  to verify the formulation, which achieved a determinant of the coefficient $R^2=0.998$. The details of the example are also demonstrated in Appendix A.4.
>
> **Question 3:** Figure 4 is not representative enough.
>
> **Answer:** Thanks a lot for the suggestions. We have visualized the loss landscape with respect to the adversarial perturbations in the CIFAR10 dataset in Figure 8 in Appendix A.5. The visualization also indicates that both SAM and ESAM improve the sharpness significantly in comparison to SGD.
>
> **Additional comments**: (5) (7) (8)
>
> **Answer:** Thank you for pointing out the writing issues. We have modified the contents in the paper as mentioned in points (5) (7) (8).

---

### Official Review · Reviewer_jSHm · 2021-11-02

**Correctness:** 3
**Technical Novelty And Significance:** 3
**Empirical Novelty And Significance:** 3
**Recommendation:** 8
**Confidence:** 4

**Main Review:**

Strengths
+ simple modifications that improves over existing SAM results
+ Wide range of baselines

Weakness
- lack of ablations across batch size (especially with the data-selection algorithm)

== On cost of SAM ==
Cost of SAM is mentioned as 100% in the text of the paper. Typical SAM application when batch sizes are larger, involve data parallel training across a flock of GPUs. The efficient version of SAM (that works in practice) do not communicate the gradients across the cores, and hence cost is not exactly 100% but lower. It would be good to know results against this standard version of SAM.

== Hyperparameter tuning ==
It would be ideal if the authors normalized for the training cost. What would be the accuracy of SGD with 40% more epochs, with appropriate tuning of learning rate schedules?

== Choice of gamma and batch size==
Is the gamma=0.5 mean 50% of examples in a batch are discarded from the gradient step? This is quite fascinating that this method was able to improve accuracy while discarding half the batch! It would be extremely valuable to know (see comment on cost of SAM) to much larger batches (for example 2048 for CIFAR10/100) if this continues to remain true.


**Summary Of The Paper:**

Paper proposes techniques to improve the efficiency of Sharpness aware minimization method. They are Stochastic Weight Perturbation (Select subset of the parameters at any step) and Sharpness-sensitive Data Selection.  Results demonstrates efficacy over SAM at small batch sizes on multiple models.



**Summary Of The Review:**

Modifications are simple and efficient in practice, and may improve the practical usage of SAM across workloads.

---

> ### Author Response · Authors · 2021-11-20
> **Response**
>
> Thanks a lot for your valuable feedback. Our responses to the weak points and questions are as below.
>
> **Question 1:** Lack of ablations across the batch size. Experiments on larger batches.
>
> **Answer:** Thank you for your suggestions; we conduct the ablation studies of batch size ranging from 1024 to 2048 and report the results as follows.
> Ablation studies with large batch sizes
>
> |ResNet-18|CIFAR-10|CIFAR-10 | CIFAR-100|CIFAR-100|
> | ----------- | :-----------: |:-----------:|:-----------: |:-----------:|
>  |Batch Size|1024 | 2048|1024 | 2048|
> |SGD | 95.35|94.72| 75.74|75.13 |
> |SAM | 96.11| 95.23 | 78.41 | 77.32|
> |ESAM(ours) | 96.26|95.74|78.64|77.71|
>
>
> |Wide-28-10 | CIFAR-10|CIFAR-10 | CIFAR-100|CIFAR-100|
> | ----------- | :-----------: |:-----------:|:-----------: |:-----------:|
> |Batch Size|1024 | 2048|1024 | 2048|
> |SGD|96.19|95.16|79.88|76.35|
> |SAM|96.80|95.69|82.84|80.88|
> |ESAM|96.98|96.03|83.13|81.19|
>
> We double the learning rate and $\rho$ and keep the rest settings the same for the batch sizes of 1024 and 2048. It can be seen that ESAM can still achieve comparable accuracy with SAM on much larger batch sizes on the CIFAR10/100 datasets.
>
>
> **Question 2:** Cost of SAM in data parallel training.
>
> **Answer:** We evaluate the training time of SAM and ESAM in the distributed data parallel (DDP) training with 2 NVIDIA 3090 GPUs. In summary, both SAM and ESAM enjoy the benefit of   around 10% training time reduction from  DDP. We report the results as follows.
>
> Training speed in DDP (images/s) on CIFAR-10
>
> |Models | ResNet-18 | Wide-28-10 | PyramidNet-110|
> | ----------- | :-----------: |:-----------:|:-----------: |
> |SGD | 7299 |1488 |880|
> |SAM|3848(100%)|782(100%)|458(100%)|
> |ESAM(ours)|5349(139.1%)|1074(137.3%)|627(136.9%)|
>
>
> **Question 3:** Normalization for the training cost
>
> **Answer:** Thank you for your constructive suggestions; we normalize the training cost of SGD(200 epochs to 280 epochs) and report them as follows.
>
> |ResNet-18 | CIFAR-10 | CIFAR-100|
> | ----------- | :-----------: |:-----------:|
> |SGD | 95.63 | 78.69 |
> |SAM | 96.51 | 80.17|
> |ESAM(ours) | 96.56 | 80.41|
>
> |Wide-28-10 |     CIFAR-10 | CIFAR-100|
> | ----------- | :-----------: |:-----------:|
> |SGD|96.56|81.98|
> |SAM|97.27|83.42|
> |ESAM(ours)|97.29|84.51|
>
> It can be seen that more training epochs of SGD does improve the accuracies on both CIFAR10 and CIFAR100 datasets. However, the improvements are marginal compared to those contributed by SAM and ESAM. We will update the normalized SGD results of the paper once we finish the normalized SGD experiments with PyramidNet-110 on the CIFAR10/100 datasets and experiments on the ImageNet dataset.
>
> **Question 4:** Choice of $\gamma$
>
> **Answer:** Yes, $\gamma=0.5$ represents 50% of the examples in the batch are discarded from the gradient step, which is the second backpropagation step used for updating the parameters. However, the discarded examples vary in each training iteration, as the instance-wise loss difference determines their selection in equation 6.

---

> > ### Comment · Reviewer_jSHm · 2021-11-25
> > **Satisfactory response from Authors**
> >
> > Authors have provide sufficient clarification, and additional results (some of which are still quite surprising to me, for example batch size filtering), I think the paper should be accepted as improvements to SAM interms of per-step-time performance is valuable to the community. My only concern is that SAM/E-SAM techniques may not be applicable on much larger datasets and is left open in this work. I am increasing my score to '8' (I am more closer to '7').
> >  '

---

### Official Review · Reviewer_mQEh · 2021-11-03

**Correctness:** 4
**Technical Novelty And Significance:** 4
**Empirical Novelty And Significance:** 4
**Recommendation:** 8
**Confidence:** 4

**Main Review:**

This is a nice paper, that is well written and addressing a very clear problem: reducing the computational complexity of SAM. It proposes two simple tricks that both make sense, and shows that they improve efficiency without hurting performance.

## SWP Implementation

I wanted to ask a question about SWP: how do you implement it in practice? I am a bit surprised that not computing the gradients for a random subset (e.g. 50%) of weights leads to a significant improvement in performance. Do you have to do any tricks to achieve this improvement?

## Alignment of SSP gradient with full-batch gradient

You mention in page 5 that the gradient on the subset selected by SSP is well aligned with the gradient on the full batch. Did you try to actually measure this alignment, e.g. in terms of cosine similarity. It could be interesting to compare it to e.g. a random subset and $\mathbb{B}^{-}$.

## Why does ESAM outperform SAM?

In some of your experiments ESAM outperforms SAM in accuracy. Do you have ideas for why this could be happening?

## ImageNet results

One concern I have with the paper is that the original SAM paper [1] seems to report better results for SAM in some of the same settings than you do. In particular, on ImageNet in Table 2 you report 76.7 for ResNet-50 with SAM and 77.05 with ESAM. However, [1] reports 77.5 with SAM in Table 2. Similarly, their results for ResNet-101 are better, and the results for PyramidNet on CIFAR are better.

Is there a difference in the setting you use?

## References

[1] Sharpness-Aware Minimization for Efficiently Improving Generalization
Pierre Foret, Ariel Kleiner, Hossein Mobahi, Behnam Neyshabur

**Summary Of The Paper:**

The paper proposes two simple modifications of the SAM optimizer that allow to significantly improve its computational efficiency without sacrificing performance.

**Summary Of The Review:**

Overall, this is a very nice paper. My only concern is about the results relative to the original SAM paper. If the authors address this concern in the rebuttal, I am happy to recommend this paper for acceptance.

---

> ### Author Response · Authors · 2021-11-20
> **Response (1/2)**
>
> We thank the reviewer for the positive assessment. We address issues raised and additional comments mentioned below.
>
> **Question 1:** SWP implementation in practice.
>
> **Answer:** We implement the SWP by randomly setting the "requires_grads" flag of kernels' parameters to false in each layer. There is no specific trick but two points to note about the implementation of SWP. (a)  A kernel's parameters tensor is the basic unit whose “requires_grads” can be set to false in the PyTorch framework. We resample these kernels randomly in each iteration. (b) In Distributed Data Parallel (DDP), the selection of parameters by SWP is the same among different GPUs. The implementation codes of SWP have been attached in the supplementary files.
>
>
>
> **Question 2:** The alignment of SDS gradients with full-batch gradient
>
> **Answer:** Yes, we have done experiments to evaluate how well the SDS gradients align will the full-batch gradient. The experiments are presented in Figure 6 in appendix A.2.  We measured the gradients' alignment by the cosine similarity between the gradients from  $\mathbb{B}^+$ and $\mathbb{B}^-$ and the gradients from the full-batch $\mathbb{B}$. . From the results,  the gradients computed by $\mathbb{B}^+$ have higher cosine similarities with the full-batch $\mathbb{B}$ than $\mathbb{B}^-$.
>
> As you suggested, we have also measured the gradients' alignment of the random subset and updated it in appendix A.2 Figure 6. We also demonstrate the gradients’ alignment experimental results with WideResNet-10 on the CIFAR100 dataset as follows.
>
> Cosine Similarity measurement of gradients with Wide-28-10 on CIFAR100
>
> |Cosine Similarity |50 |100|150 | 200 |
> | ----------- | :-----------: |:-----------: |:-----------: |:-----------: |
> | CosSim($\nabla_\theta L_{\mathbb{B}^-}(f_{\theta+\hat{\epsilon}}),\nabla_\theta L_{\mathbb{B}}(f_{\theta+\hat{\epsilon}})$) | 0.192 |0.132|0.077 |0.072 |
> |CosSim($\nabla_\theta L_{\mathbb{B}\_{rand}}(f_{\theta+\hat{\epsilon}}),\nabla_\theta L_{\mathbb{B}}(f_{\theta+\hat{\epsilon}})$) |0.727|0.699|0.696|0.707|
> |  CosSim($\nabla_\theta L_{\mathbb{B}^+}(f_{\theta+\hat{\epsilon}}),\nabla_\theta L_{\mathbb{B}}(f_{\theta+\hat{\epsilon}})$) | 0.938 | 0.933 | 0.930| 0.926|
>
>
>
>
> From the experimental results in the above table and Figure 5 of appendix A.2, the gradients computed by $\mathbb{B}^+$ have much higher cosine similarities with the full-batch $\mathbb{B}$ than $\mathbb{B}^-$ and the random set. The results verify the effectiveness of the SDS.
>
> **Question 3:** The reason that ESAM outperforms SAM.
>
> **Answer:** Here we provide a discussion about the accuracy improvement contributed by ESAM.  A plausible reason for such improvement is that SWP leads to a better inner maximum in equation 2. The current approximate solution $\hat{\epsilon}$ is obtained based on the assumption that  $L_{\mathbb{B}}(f_{\theta})$ is a linear function. SWP improves the linearity of $L_{\mathbb{B}}(f_{\theta})$ and thus leads to a more accurate solution $\hat{\epsilon}$. We prove it empirically as follows. Inspired by [4], we measure the linearity of the loss function by
>
> $\zeta(\epsilon,\mathbb{B}) = |L_{\mathbb{B}}(f_{\theta+\epsilon})-L_{\mathbb{B}}(f_{\theta})-\epsilon^{\top}\nabla_{\theta} L_{\mathbb{B}}(f_{\theta})  |$
>
> where $\epsilon$ is the weight perturbation with a small magnitude. We measure the linearity of SWP with ResNet-18 on the CIFAR10 dataset. The smaller value of $\zeta(\epsilon,\mathbb{B})$ indicates better linearity. The experimental results are demonstrated as Figure 7 in appendix A.4, which verify that SWP improves the linearity of $L_{\mathbb{B}}(f_{\theta})$.
>
> The reason that SDS improves the accuracy is as indicated by Equation (10) in section 2.3. We demonstrate that SDS can select a subset $\mathbb{B}^+$ that results in a better upper bound of SAM loss. We also verify Equation (10) empirically by the experiments shown in Figure 5.

---

> > ### Author Response · Authors · 2021-11-20
> > **Response (2/2)**
> >
> > **Question 4:** ImageNet results.
> >
> > **Answer:** Yes, as the SAM paper [1] does not introduce the exact experimental settings for the PyramidNet on CIFAR10/100 and ImageNet experiments, we follow the experimental settings of different papers [2][3]. For imageNet experiments, SAM paper[1] does not introduce the settings such as the value of $\rho$, learning rate, weight decay,  and data augmentations. These experimental settings will affect the accuracy seriously. In particular, the settings of data augmentations should be consistent for a fair comparison. Therefore, we follow the experimental settings in [2], which applies the SAM on ImageNet and report the experimental settings clearly. Our results are the same as the results reported in [2].
> >
> > Moreover, the architectures of PyramidNet are not specified clearly in the SAM paper [1] as well. For example, the settings of $\alpha$, depth, and w.t./w.o. bottleneck have not been introduced in the SAM paper [1]. Therefore, we refer to the original paper of PyramidNet [3] and follow the settings the same as [3] suggested. We choose PyramidNet-110 with $alpha=270$ depth$=110$ for evaluation. Our SGD results are the same as those reported in [3] (ours 81.89% v.s. 81.75% in [3]).
> >
> >
> > References:
> >
> > [1] Pierre Foret, Ariel Kleiner, Hossein Mobahi, and Behnam Neyshabur. Sharpness-aware minimization
> > for efficiently improving generalization. arXiv preprint arXiv:2010.01412, 2020.
> >
> > [2] Xiangning Chen, Cho-Jui Hsieh, and Boqing Gong. When vision transformers outperform resnets
> > without pretraining or strong data augmentations. arXiv preprint arXiv:2106.01548, 2021.
> >
> > [3] Dongyoon Han, Jiwhan Kim, and Junmo Kim. Deep pyramidal residual networks. In Proceedings
> > of the IEEE conference on computer vision and pattern recognition, pp. 5927–5935, 2017. https://github.com/dyhan0920/PyramidNet-PyTorch
> >
> > [4]Qin, Chongli, et al. "Adversarial robustness through local linearization." arXiv preprint arXiv:1907.02610 (2019).

---

> > > ### Comment · Reviewer_mQEh · 2021-11-29
> > > **Thank you for the detailed rebuttal**
> > >
> > > Dear authors, thank you for the detailed rebuttal, especially for the new experimental results and intuition for why ESAM is helpful! I maintain my assessment, this is a good paper and I recommend accepting it.

---

### Author Response · Authors · 2021-11-20
**Summary of changes**

Dear reviewers:

We would like to thank all the reviewers for the constructive suggestions on our paper. We have updated the paper as follows to address the reviewers’ comments.

1. We have modified the typos and writing issues as suggested by reviewers.

2. **We have updated Figure 6 in appendix A.2 to compare the gradients' alignment** of the subset $\mathbb{B}^+$ selected by SDS to a random set in terms of cosine similarity.  The results demonstrated that the gradients computed by $\mathbb{B}^+$ have much higher cosine similarities with the full-batch $\mathbb{B}$ than $\mathbb{B}^-$ and the random set.

3. **We have added the subsection A.3 to empirically investigate the reason that SWP can improve the accuracy of ESAM.** The experimental results indicate that SWP can improve the linearity of loss function and thus solve the inner maximization problem better in Equation (2).

4. **We have added the subsection A.4 to formulate the reduced computational overhead contributed by SWP.** The saved computational overhead is mainly resulted from the storing and hooking gradients of the parameters that have not been sampled by SWP. In a nutshell, the reduced training time contributed by SWP is approximately linear with $\beta$ and $\frac{1-\beta}{N\beta}$, where N is the depth of neural networks. We also conducted a toy example to verify the formulation, which achieved a $R^2=0.998$.

5. **We have added the subsection A.5 to visualize the loss landscape with respect to the adversarial perturbations in the CIFAR10 dataset.** The visualization indicates that both SAM and ESAM improve the sharpness significantly in comparison to SGD.

6. **We have added experiments in the subsection A.6 to evaluate the performance of ESAM on the new ViT-S/16 architecture.**

---

### Public Comment · ~Fabian_Latorre1 · 2023-03-10
**Please clarify how the hyperparameters were chosen**

Dear authors,

could you please clarify how did you choose the final learning rate for the numbers reported? From page 6 in your paper you claim
> Additionally, the other hyperparameters of SGD, SAM and ESAM have been tuned separately for optimal test accuracies using grid search.

Does this mean that you chose the parameters that returned the highest **test** accuracies and that is the value you report? If that is the case the reported numbers are not an unbiased estimate of the true generalization error and hence, they are not meaningful.

Hopefully this can be corrected or clarified

Best,

---

> ### Public Comment · ~Jiawei_Du1 · 2023-03-13
> **hyperparameters chosen**
>
> Dear Fabian Latorre,
>          Thanks for your question. We follow the experimental setting of [chen et al. 2021] to do the grid search on the validation set. To ensure a fair comparion, we reproduce the SAM's result as reported in [chen et al. 2021], and keep all the hyperparameters the same between SAM and ESAM as reported in Table 8 of our paper to compare their performance and effciency.
> Best regards
> Jiawei

---

### Decision · Program_Chairs · 2022-01-20

**Decision:**

Accept (Poster)

**Comment:**

This paper focuses on improving the efficiency of sharpness-aware minimization method for training neural networks. The proposals are stochastic weight perturbation, namely selecting subset of the parameters at any step, and sharpness-sensitive data selection. The philosophy behind sounds quite interesting to me, namely, sharpness-aware minimizer can be approximated properly with fewer computations after analyzing the min-max procedure. This philosophy leads to a novel algorithm design I have never seen.

The clarity and novelty are clearly above the bar of ICLR. While the reviewers had some concerns on the significance, the authors did a particularly good job in their rebuttal. Thus, all of us have agreed to accept this paper for publication! Please include the additional experimental results in the next version.